# New retron systems from environmental bacteria identify triggers of anti-phage defense and expand tools for genome editing

**Kazuo L. Nakamura**[1,2◉]**, Karen Zhang**[1,3*◉]**, Mario R. Mestre**[4]**, Matías Rojas-Montero**[1]**,
Seth L. Shipman** [1,5,6*]

**1** Gladstone Institute of Data Science and Biotechnology, San Francisco, California, United States of America, **2** Undergraduate Program in Bioengineering, University of California, Berkeley, California, United States of America, **3** Graduate Program in Bioengineering, University of California, San Francisco and Berkeley, San Francisco, California, United States of America, **4** Section of Microbiology, Department of Biology, University of Copenhagen, Copenhagen, Denmark, **5** Department of Bioengineering and Therapeutic Sciences, University of California, San Francisco, California, United States of America, **6** Chan Zuckerberg Biohub, San Francisco, California, United States of America

◉ These authors contributed equally to this work.
\* karen.zhang@gladstone.ucsf.edu (KZ); seth.shipman@gladstone.ucsf.edu (SLS)

## Abstract

Retrons are bacterial immune systems that protect a bacterial population against phages by killing infected hosts. Retrons typically comprise a reverse transcriptase (RT), a template noncoding RNA that is partially reverse transcribed into RT-DNA, and a toxic effector. The reverse transcriptase (RT), noncoding RNA, and RT-DNA complex sequester the toxic effector until triggered by phage infection, at which point the toxin is released to induce cell death. Due to their ability to produce single-stranded DNA in vivo, retrons have also been engineered to produce donor templates for genome editing in both prokaryotes and eukaryotes. However, the current repertoire of experimentally characterized retrons is limited, with most retrons sourced from clinical and laboratory strains of bacteria. To better understand retron biology and natural diversity, and to expand the current toolbox of retron-based genome editors, we developed a pipeline to isolate retrons and their bacterial hosts from a variety of environmental samples. Here, we identify seven new retron systems, each isolated from a different host bacterium. We characterize DNA production by these retrons and test their ability to defend against a panel of *Escherichia coli* phages. We find that two of these retrons are disrupted by other elements, in one case a group II intron and in another a separate defense system, yet both retrons still produce RT-DNA. For two other retrons, we further unravel their mechanism of defense by identifying the phage genes responsible for triggering abortive infection. Finally, we engineer these retrons for genome editing in *E. coli*, demonstrating their potential use in a biotechnological application.

**Data availability statement:** Sequencing data associated with this study are available in the NCBI SRA (PRJNA1196507) https://www.ncbi.nlm.nih.gov/sra/PRJNA1196507.

**Funding:** Work was supported by awards and funding to S.L.S from the National Science Foundation (MCB 2509382, https://www.nsf.gov/bio/mcb), the Robert J Kleberg, Jr. and Helen C. Kleberg Foundation (https://www.klebergfoundation.org/), and the Gary and Eileen Morgenthaler Fund. Additional funding from the Gordon and Betty Moore Foundation (https://moore.org/). S.L.S. is a Chan Zuckerberg Biohub—San Francisco Investigator (https://www.czbiohub.org/sf/). K.Z. was supported by a National Science Foundation Graduate Research Fellowship (https://www.nsf.gov/) and a UCSF Discovery Fellowship (https://graduate.ucsf.edu/admission/financial-support/fellowships/discovery-fellows-program). The funders had no role in study design, data collection and analysis, decision to publish, or preparation of the manuscript.

**Competing interests:** The authors declare no competing interests.

**Abbreviations:** ncRNA, noncoding RNA; ONT, Oxford Nanopore Technologies; PADLOC, Prokaryotic Antiviral Defense Locator; PDC, phage defense candidate; RT, reverse transcriptase; RT-DNA, reverse-transcribed DNA; ssDNA, single-stranded DNA; TdT, terminal deoxynucleotidyl transferase.

## Introduction

Retrons are multicomponent bacterial immune systems that trigger abortive infection in response to phage. The components of these systems include a reverse transcriptase (RT), a noncoding RNA (ncRNA) that is partially reverse transcribed to a hybrid RNA/DNA molecule, and one or more effector proteins that typically function as toxins to the retron host [1–3]. The toxicity of the effector protein(s) is neutralized in the presence of the RT and reverse-transcribed DNA (RT-DNA), but perturbation of the RT-DNA by phage-encoded elements can release this neutralization, leading to abortive infection. Beyond their natural function in phage defense, retrons have also been repurposed into components of biotechnology [4–6]. For instance, retron ncRNA can be modified so that reverse transcription produces template DNA for genome editing [7–18].

Retrons were initially discovered in Myxobacteria when researchers noticed unexplained bands running on acrylamide gels of ~120 nucleotides in length from total nucleic acid preparations [19]. Subsequent studies determined that these bands were single-stranded DNA (ssDNA), produced by a retron RT from a retron ncRNA [20]. More recently, thousands of retrons have been bioinformatically predicted in diverse bacterial species and categorized into clades and types [21]. These retrons have different effector proteins and mechanisms, as well as widely varying RT-DNA production and utility in gene editing [22].

Despite this, only a handful of retrons have been studied in the context of their natural host, and retron defense against phages has only been studied in *Escherichia coli* [3,23–25], *Bacillus subtilis* [26], and *Salmonella enterica* [1]. Identifying a wider range of natural hosts would enable researchers to determine whether mechanisms identified biochemically or within model systems like *E. coli* accurately represent their biology. Furthermore, prior studies have been largely limited to retrons from clinical and laboratory strains of bacteria with few environmental examples (e.g., *Vibrio mimicus* [27]). Given the limited selection of experimentally confirmed retrons, characterization of additional retrons sourced from different environments could lead to the discovery of new defense mechanisms and the development of better biotechnology for application in molecular genetics.

In this study, we develop an approach to identify novel retrons in bacteria isolated from soil and water samples by screening for RT-DNA bands using DNA isolation and polyacrylamide gel electrophoresis. Here, we add seven new environmental retron hosts. We demonstrate that six of the seven naturally sourced retrons can be ported into *E. coli* to produce RT-DNA and two exhibit defense against *E. coli* phages. By sequencing phage escapees, we determine that the Bas51 N6 adenine methyltransferase is a trigger for these two retrons and extend the mechanism to show that this trigger activates growth suppression in the natural host of one retron. We also demonstrate that these environmentally sourced retrons can be used for genome editing in *E. coli*. This process for isolating and identifying novel retrons can be used to expand our understanding of retron biology by providing more natural hosts for study and aid in the development of new tools for genome editing.

# Results

## Isolation of retron-bearing hosts from environmental samples

To isolate new retron hosts from the environment, soil and water samples were collected from a range of climates and environments (S1 Table). Soil samples were added to LB media and broken down by vortexing with beads. Rocks and soil debris were removed by centrifugation through a filter. The resulting media containing soil microbes was plated on LB agar. Colonies were picked and grown for 16 h in LB media, then passaged at a 1:100 dilution and grown for an additional 7 h in LB media (Fig 1a). Water samples were plated directly, and colonies were subcultured using the steps

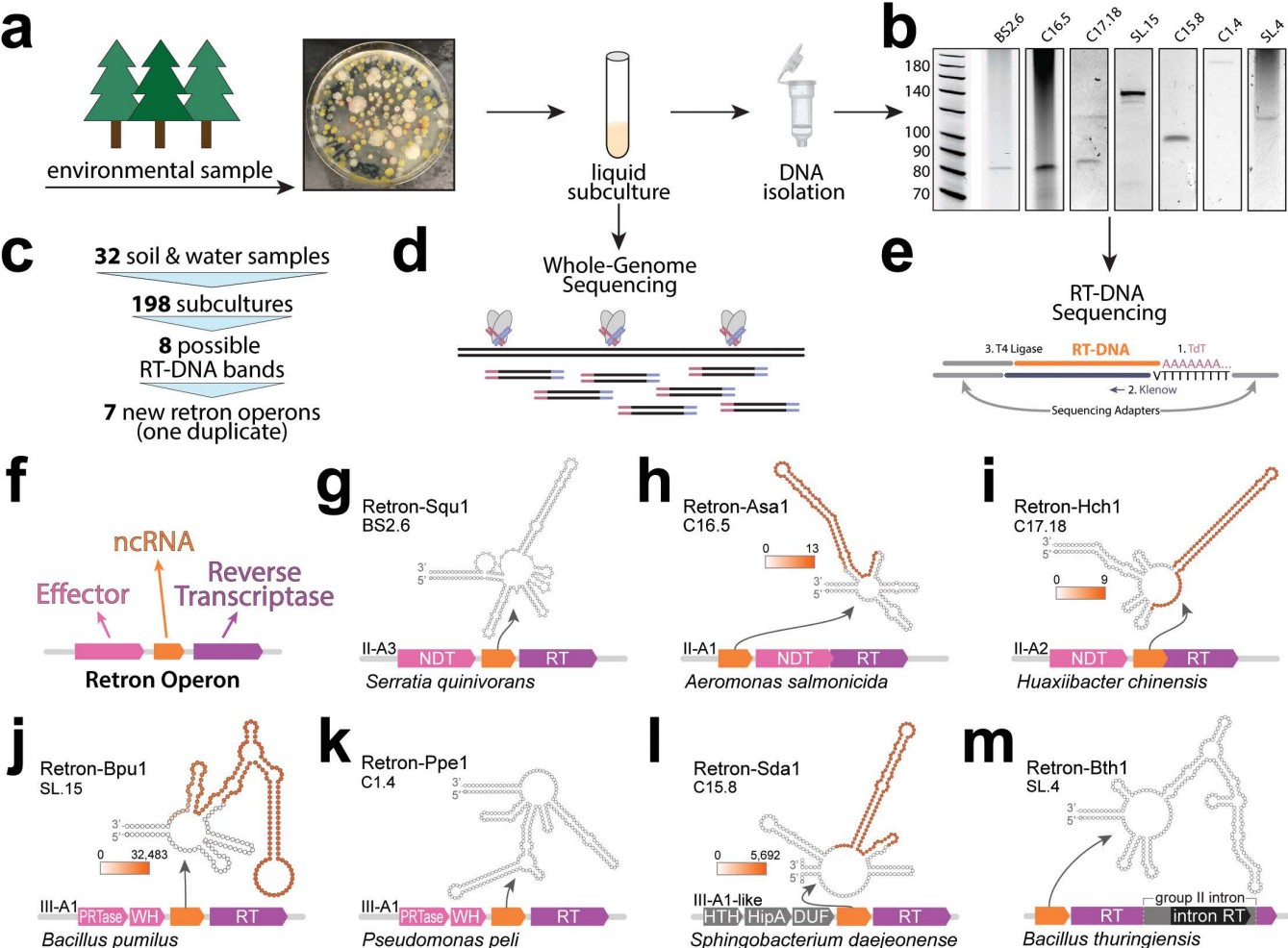

**Fig 1. Isolation of retron-bearing hosts from environmental samples. a)** Schematic of the isolation of bacteria from soil and water samples and identification of possible retron-bearing hosts using PAGE analysis. **b)** Retron RT-DNA bands from PAGE analyses (uncropped gels included as S1 Raw Images). **c)** Overview of sampling, screening of colonies, and identification of retron hosts. **d)** Hosts displaying an RT-DNA band were subjected to whole-genome sequencing. **e)** RT-DNA was subjected to unbiased adapter addition and sequencing. **f)** Schematic of canonical retron operon with components annotated, effector in pink, noncoding RNA (ncRNA) in orange, and reverse transcriptase in purple. f–m) Seven putative retron operons, identifying the retron type, host species, and predicted two-dimensional structure of the ncRNA. For retrons where RT-DNA sequencing is available, sequencing coverage is plotted onto ncRNA in an orange heatmap (heatmap legend indicated number of sequencing reads mapped). Gene abbreviations: NDT (nucleoside deoxyribosyltransferase-like); PRTase (phosphoribosyltransferase); WH (winged helix-turn-helix DNA binding); HTH (helix-turn-helix domain containing); HipA (HipA-like); DUF (DUF3037 domain-containing protein).

described above. All samples in this study were initially cultured at 37 °C. Two samples, later identified as *Serratia quinivorans* and *Huaxiibacter chinensis*, were cultured at 30 °C in subsequent experiments after exhibiting faster growth at this temperature.

We identified retron-bearing hosts by the presence of the characteristic retron ssDNA that runs at 40–180 nucleotides on a polyacrylamide gel. To do so, we prepped total DNA from the 7 h cultures using Qiagen miniprep kits and ran this DNA on TBE-Urea gels (Fig 1b). In total, 13 water samples and 19 soil samples were collected. DNA was isolated from 198 subcultures. Twenty-nine of the subcultures exhibited possible bands on initial gels. Seven subcultures were not followed up because they duplicated band positions from other bacteria originating from the same environmental sample. The remaining 22 subcultures with possible bands were recultured and rechecked. Eight of these subcultures could not be recovered from glycerol stocks and six subcultures showed no distinct bands upon rechecking, leaving eight subcultures with reliable RT-DNA production. We sequenced the genomes of the remaining eight subcultures, which revealed seven putative retron operons with one duplicate strain (Fig 1c and S3 Table).

Bacterial subcultures showing a putative retron RT-DNA band underwent both whole genome sequencing, as well as ssDNA isolation followed by RT-DNA sequencing (Fig 1d and 1e). Whole genome data was analyzed using the Prokaryotic Antiviral Defense Locator (PADLOC) [28] tool, which identifies retron operons and the genes encoding the effector protein, ncRNA, and RT, which are present in most retron types (Fig 1f). We also analyzed whole genome contigs using PubMLST, which identified the species of each bacterial host based on genes encoding ribosome protein subunits. In total, PADLOC predicted novel retron operons in the genomes of five of the seven potential hosts, with a lone retron RT also identified in *Sphingobacterium daejeonense*. Another lone retron RT was found in *Bacillus thuringiensis* by manually checking motifs identified in the HMMERS section of the DefenseFinder [29] tool and by searching broadly for RTs using MyRT [30]. Retrons have been classified into 13 types based on effector function and operon structure [21]. Five of the natural operons identified here could be categorized into known retron types: II-A for *S. quinivorans* (Retron-Squ1), *Aeromonas salmonicida* (Retron-Asa1), and *H. chinensis* (Retron-Hch1); and III-A for *Bacillus pumilus* (Retron-Bpu1) and *Pseudomonas peli* (Retron-Ppe1) (Fig 1g–1k). The remaining two putative retrons did not fall into a previously characterized type. The putative retron in *Sphingobacterium daejeonense* (Retron-Sda1) contained a III-A1-like RT, but without the PRTase/WH effector genes that are typical of III-A1 (Fig 1l). The putative retron in *Bacillus thuringiensis* (Retron-Bth1), which was mapped to an extrachromosomal element rather than the bacterial genome, had no obvious effector, and contained a group II intron, a self-splicing mobile genetic element, within the coding sequence of the putative retron RT (Fig 1m). Ignoring the group II intron, the Retron-Bth1 RT is unclassified among other retron RTs. For each retron operon, we identified a characteristic retron ncRNA based on secondary structure. In four cases, we were able to sequence the retron RT-DNA and plot coverage onto the ncRNA (coverage in orange).

## Naturally sourced retrons can produce RT-DNA in *E. coli*

To validate that the retron systems we identified were complete, we amplified them from the genomes of their natural hosts, cloned them onto plasmids, and transformed them for expression in *E. coli* bSLS114 (a derivative of BL21-AI with endogenous Retron-Eco1 removed [4]) downstream of an inducible T7/lac promoter (Fig 2a). In cloning retron operons, we included 100–150 bases before the start codon of the first gene to capture the native promoter.

We analyzed RT-DNA production in both uninduced and induced conditions, to see if the systems were competent for production and if the systems carried their own constitutive promoters that are active in *E. coli*. Similar to analysis from within the natural host, cultures were grown at 37 °C in LB media overnight, then passaged at a 1:100 dilution and grown for an additional 7 h the following day with a previously characterized retron, Retron-Eco1, run in parallel as a control for RT-DNA production. DNA was isolated using a Qiagen miniprep then analyzed on a TBE-Urea gel (Fig 2b).

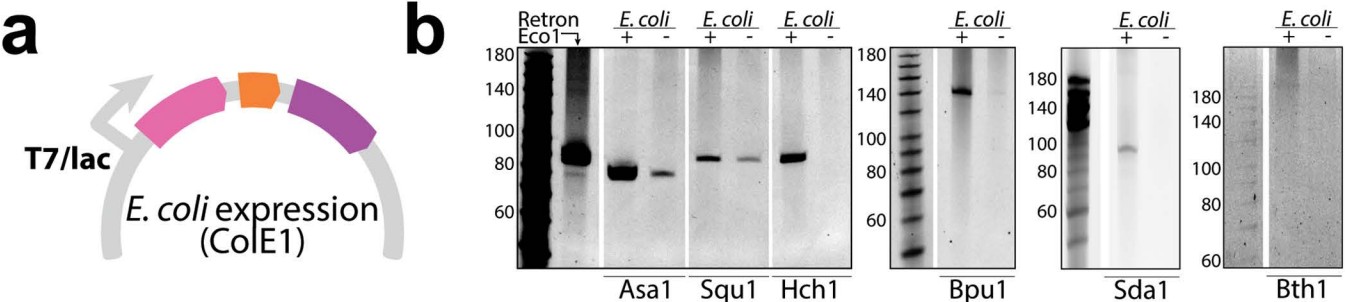

**Fig 2. Naturally sourced retrons can produce RT-DNA in *E. coli*. a)** Schematic of plasmid for retron expression in *E. coli*. **b)** Images from PAGE analyses of RT-DNA production from naturally sourced retrons in induced (+) and uninduced (−) conditions (uncropped gels included as S1 Raw Images). Lengths for a single-stranded ladder are indicated to the left of each gel and Retron-Eco1 is shown in the first lane for comparison.

Retron-Ppe1, identified in *Pseudomonas*, was not studied in *E. coli* because it could not be transformed, even in the presence of 1% glucose as a repressor of the inducible promoter. We presume that this retron is constitutively expressed and toxic to *E. coli*, perhaps due to the presence of a trigger for this retron in *E. coli*, or the absence of a neutralizing factor from the natural host.

Every other retron produced RT-DNA in *E. coli* when overexpressed from the T7 promoter (with induction). Retrons-Asa1, Squ1, and Bpu1 additionally produced detectable RT-DNA without induction suggesting that their endogenous promoters are sufficient for constitutive expression in *E. coli*. This data confirms that the components in each putative retron operon are sufficient for RT-DNA production. RT-DNA length exactly matches the length initially observed for all retrons other than Retron-Bth1, further indicating that these operons contain the elements responsible for the RT-DNA in the natural hosts. The RT-DNA produced by the Retron-Bth1 operon in *E. coli* is longer than the band observed in the natural host. While this could indicate that a different element is responsible for RT-DNA production in the natural host, we found no other candidate. Thus, we speculate that the longer RT-DNA is likely due to a difference in termination or degradation of the RT-DNA in the host versus *E. coli*.

## Retrons exhibit phage defense

To better understand how naturally sourced retrons function as defense systems, we next tested the four retrons with intact, type-identified operons for defense against a diverse panel of 12 *E. coli* phages, including examples of *Drexlerviridae* (Bas01), *Siphoviridae* (Bas15, Bas20, Bas21), *Demerecviridae* (T5), *Myoviridae* (Bas37, T4, P1), *Mosigvirus* (Bas46), *Vequintavirinae* (Bas51), *Autographiviridae* (T7), and *Siphoviridae* (lambda) [31]. The more complex cases of retrons-Sda1 and -Bth1 are experimentally addressed in the next section. For this test, cultures of *E. coli* MG1655 were transformed with Retrons-Asa1, -Squ1, -Hch1, and -Bpu1 (same plasmids as Fig 2), first using the endogenous promoters to drive retron expression. Transformed strains were plated in LB agar. Phages were then titrated and spotted, and plates were left at 37 °C overnight for plaques to form.

We found that Retron-Asa1 exhibited defense against four phages: Bas01, Bas20, Bas21, and Bas51 (Fig 3a) and Retron-Squ1 exhibited defense against six phages: Bas01, Bas15, Bas20, Bas21, Bas51, and T5 (Fig 3b). Retrons-Hch1 and -Bpu1 exhibited no defense phenotype against any of the phages tested. One potential explanation for a lack of phage defense from retrons-Hch1 and -Bpu1 could be weak expression from their endogenous promoters in *E. coli*. Indeed, the two retrons that did defend produced more RT-DNA in uninduced conditions than the two retrons that did not show a defense phenotype (Fig 2b). To check, we tested for phage defense against the same panel of phages with retrons-Hch1 and -Bpu1 overexpressed from a T7 promoter in strain bSLS114, but we still found no defense phenotypes for these retrons against this phage panel (S1 Fig).

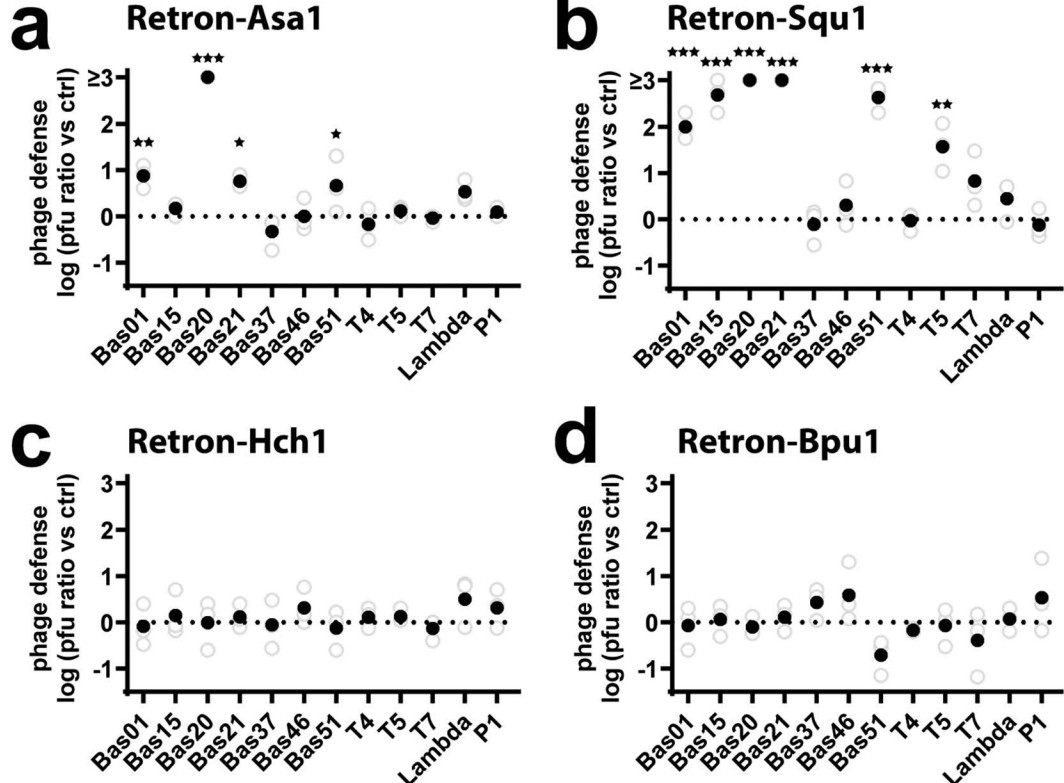

**Fig 3. Retrons-Asa1 and -Squ1 Exhibit Defense. a)** Phage defense (log transformed) was calculated as the pfu (plaque-forming units) per ml on a control strain not expressing the retron divided by pfu per ml on the retron-expressing strain. Increasing numbers indicate defense phenotypes. Three biological replicates are shown as open circles and the mean is shown as a closed circle. Retron-Asa1 exhibits defense (one-way ANOVA $P < 0.0001$) against Bas01 ($P < 0.0038$), Bas20 ($P < 0.0001$), Bas21 ($P < 0.0141$), and Bas51 ($P < 0.0381$). Individual phage statistics calculated by Dunnett's test corrected for multiple comparisons. **b)** Phage defense shown exactly as in A, but for Retron-Squ1, which exhibits defense (one-way ANOVA $P < 0.0001$) against Bas01 ($P < 0.0001$), Bas15 ($P < 0.0001$), Bas20 ($P < 0.0001$), Bas21 ($P < 0.0001$), Bas51 ($P < 0.0001$), and T5 ($P = 0.0001$). Individual phage statistics calculated by Dunnett's test corrected for multiple comparisons. **c)** Phage defense shown exactly as in A, but for Retron-Hch1, which does not exhibit defense against this panel of phages (one-way ANOVA $P = 0.7125$). **d)** Phage defense shown exactly as in A, but for Retron-Bpu1, which does not exhibit defense against this panel of phages (one-way ANOVA $P = 0.0644$). Additional statistical details in S3 Table. The data underlying all panels in this figure can be found in S4 Table.

## Defense mechanisms of Retron-Asa1 and -Squ1

We next identified the triggering mechanism for Retrons-Asa1 and -Squ1, using Bas51 phage. Cells transformed with retrons as above were plated and Bas51 was spotted at low concentrations, allowing phages with mutations in genes triggering the retron to escape retron defense and form plaques.

We used nanopore sequencing to identify potential trigger genes in mutant phage escapees, sequencing four escapees of Retron-Asa1 and four of Retron-Squ1. When analyzing data, we ignored mutations in homopolymeric regions (known to be enriched for nanopore errors [32]) and synonymous mutations. All escapees that we sequenced contained mutations in an adenine methyltransferase gene, with five distinct mutations across the eight escapee phages (Fig 4a). Additionally, four of eight contained an identical premature stop mutation in a glycosidase gene (Fig 4b).

To determine whether either the methyltransferase or glycosidase gene functioned as a trigger, we initially tried expressing the genes individually in the presence or absence of the retrons to see if either resulted in a growth defect in the presence of the retron. However, we found that the methyltransferase was toxic to cells even in the absence of the retron.

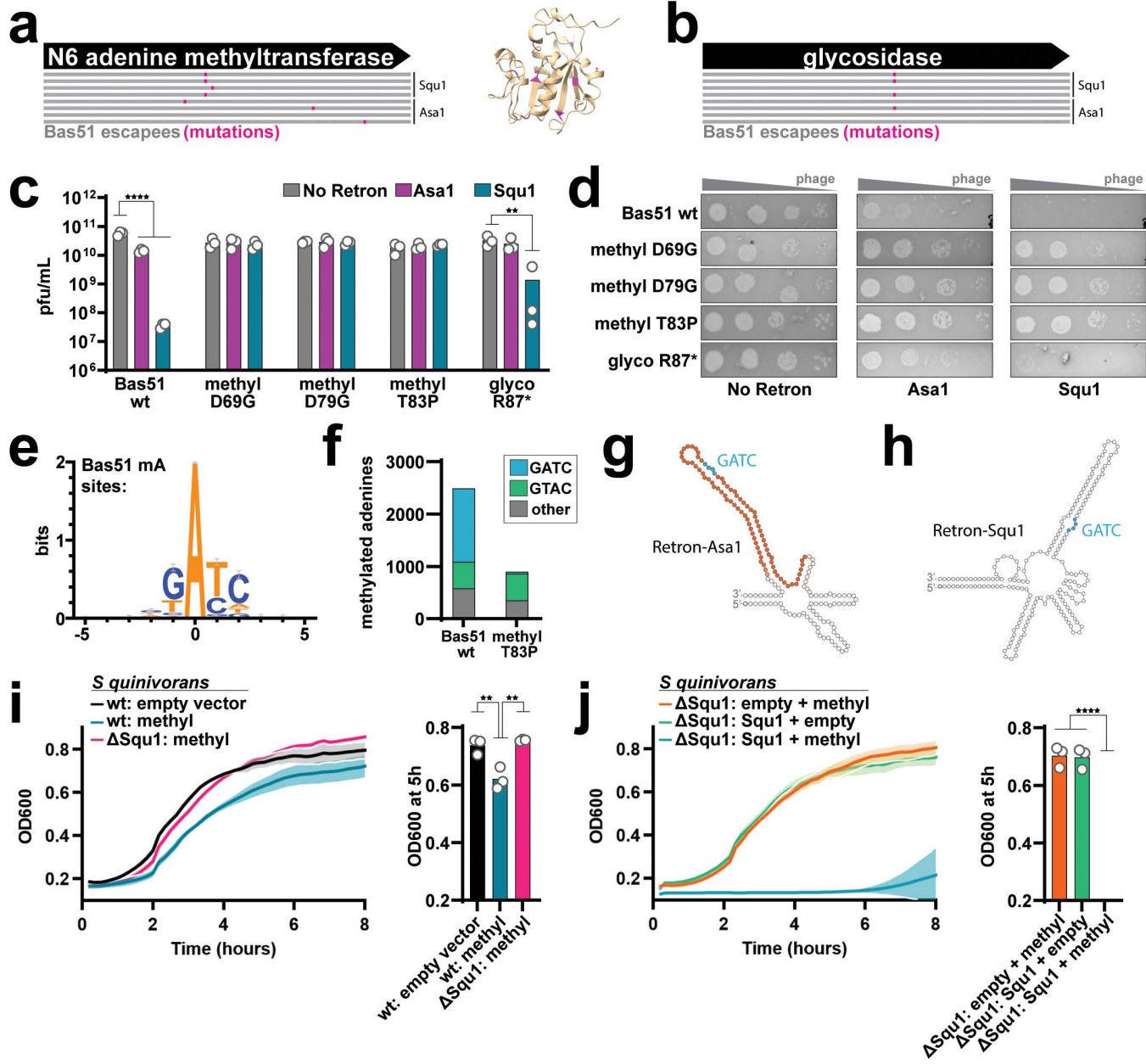

**Fig 4. Defense Mechanisms of Retron-Asa1 and -Squ1. a)** Left, phage escapee mutations in Bas51 N6 adenine methyltransferase gene on strains containing Asa1 and Squ1 constructs; right, predicted 3D structure (AlphaFold3) of methyltransferase with mutations in pink. **b)** Phage escapee mutations in Bas51 glycosidase gene on strains containing Asa1 and Squ1 constructs. **c)** PFU/mL of Bas51 phage with mutations in methyltransferase and glycosidase genes on strains containing Asa1 and Squ1 constructs (Two-way ANOVA effect of retron $P<0.0001$, phage $P=0.1244$, and interaction $P<0.0001$; follow-up testing with Tukey's, corrected for multiple comparisons, wild-type Bas51 no retron vs. Retron-Asa1 $P<0.0001$ and -Squ1 $P<0.0001$, glyco R87* no retron vs. Squ1 $P=0.0001$, all other conditions ns. The data underlying this panel can be found in S4 Table. **d)** Titration of Bas51 phage with mutations in methyltransferase and glycosidase genes on strains containing Retron-Asa1 and -Squ1 constructs. **e)** Logo showing the predominant motif of methylated adenine sites in wild-type Bas51 phage. **f)** Stacked bar plot showing distribution of adenine methylation site motifs in wild-type Bas51 phage compared to a Bas51 methyltransferase mutant (T83P). The data underlying this panel can be found in S4 Table. **g)** Retron-Asa1 predicted ncRNA secondary structure with reverse transcribed region colored in orange and GATC motif colored in blue. **h)** Retron-Squ1 predicted ncRNA secondary structure with GATC motif colored in blue. **i)** Growth curves comparing $OD_{600}$ of wild-type *Serratia quinivorans* transformed with an empty vector, wild-type *S. quinivorans* transformed with methyltransferase, and *S. quinivorans* ΔSqu1 transformed with methyltransferase over 8 h. Methyltransferase expression was induced at 2 h. Bar plot on right compares $OD_{600}$ measurements of these samples at the 5 h time point. Open circles show each of three biological replicates and the bar shows the mean (One-way ANOVA $P=0.0017$; follow-up testing with Tukey's, corrected for

multiple comparisons, wt:empty vs. wt:methyltransferase $P = 0.0045$; wt:methyltransferase vs. ΔSqu1:methyltransferase $P = 0.0021$, all other conditions ns). The data underlying this panel can be found in S4 Table. **j)** Growth curves comparing $OD_{600}$ of *S. quinivorans* ΔSqu1 transformed with either empty vector + methyltransferase, Retron-Squ1 + empty vector, or Retron-Squ1 + methyltransferase over 8 h. Methyltransferase expression was induced at 2 h. Bar plot on right compares $OD_{600}$ measurements of these samples at the 5 h time point. Open circles show each of three biological replicates and the bar shows the mean. (One-way ANOVA $P < 0.0001$; follow-up testing with Tukey's, corrected for multiple comparisons, empty&methyltransferase vs. Squ1&methyltransferase $P < 0.0001$; Squ1&empty vs. Squ1&methyltransferase $P < 0.0001$, all other conditions ns). The data underlying this panel can be found in S4 Table. For all data panels, additional statistical details in S3 Table.

To avoid the issue of methyltransferase toxicity, we instead introduced mutations in the methyltransferase and glycosidase genes in Bas51. By making these mutations precisely on a wild-type phage background, we could disentangle potential effects of the methyltransferase and glycosidase (which were often both mutated in escapees) and remove the confounding influence of other mutations that were present in escapees. In separate phages, we precisely introduced three of the escapee mutations in the N6 adenine methyltransferase (D69G, D79G, and T83P) and the premature stop mutation in the glycosidase using a recombitron approach [33].

Mutating the methyltransferase gene at any of three tested positions eliminated defense by both Retron-Asa1 and -Squ1, resulting in the same level of plaque formation as a control case expressing no retron (Fig 4c and 4d). This indicates that the Bas51 N6 adenine methyltransferase gene triggers both retrons. Introducing a premature stop mutation in the glycosidase gene also inhibited the phage defense phenotype of Retron-Asa1, but not Retron-Squ1 (Fig 4c and 4d).

We decided to further investigate the mechanism of methyltransferase-based triggering by first identifying the associated methylation site motif. We reasoned that a phage-encoded methyltransferase likely acts on the phage genome, thus we performed native sequencing of the Bas51 genome on the nanopore using a base calling algorithm that identifies N6-methylated adenines. Sequence alignment of all sites containing these methylated adenines reveals a distinct GATC motif (Fig 4e). The association of this motif with this specific methyltransferase was validated upon sequencing a Bas51 phage containing a T83P mutation in the gene, which resulted in near complete depletion of methylated adenines at GATC sites (Figs 4f and S4). This experiment additionally confirms that this mutation, which was first identified in a phage escapee, confers loss-of-function to the methyltransferase. Importantly, the GATC motif is present on both the Retron-Asa1 and -Squ1 ncRNAs and is confirmed to be reverse transcribed for Retron-Asa1 (Fig 4g and 4h).

After establishing that these retrons are triggered by a methyltransferase in *E. coli*, we wondered whether they exhibit the same behavior in their native hosts. For this, we chose to examine Retron-Squ1 in *S. quinivorans* due to the genetic tractability of the host strain. We expressed the methyltransferase in both wild-type *S. quinivorans* (which contains a genomic copy of Retron-Squ1) and *S. quinivorans* with Retron-Squ1 deleted (ΔSqu1), then measured growth of the cultures over eight hours. The wild-type strain showed reduced growth when the methyltransferase was expressed as compared to an empty vector control, and this growth effect was eliminated by deleting the endogenous Retron-Squ1 (ΔSqu1), demonstrating that this methyltransferase is able to trigger a toxic effect via the retron in its natural host (Fig 4i). Furthermore, when Retron-Squ1 borne on a high-copy plasmid was co-expressed with the methyltransferase in the ΔSqu1 strain, growth was reduced to an even greater level compared to when either plasmid was replaced with an empty vector control (Fig 4j). These results demonstrate Retron-Squ1 triggering dynamics in the context of its native host.

## Retrons-Sda1 and -Bth1 are disrupted by intervening genes

Unlike the retrons characterized above, Retrons-Sda1 and -Bth1 could not be easily categorized into existing retron types. In the case of Retron-Sda1, the identified ncRNA was classified as a type III-A1 ncRNA. However, instead of being co-located with the PRTase and DNA-binding domain-containing proteins that characterize retrons type III, Retron-Sda1 was co-located with two genes encoding proteins for HipA-like kinase and DUF3037 domains, respectively (Fig 1l). Given the lack of the usual Type III effectors, we decided to investigate this case in more depth. For this, we queried the NCBI nr

database in search of homologs for both the RT and the DUF3037 proteins, performed multiple sequence alignments and phylogenetic analyses, and retrieved the genomic neighborhoods using the NCBI Entrez API and annotated the defense systems using PADLOC. Interestingly, DUF3037 was found to be strongly associated with HipA-like containing proteins (S1 File). This two-gene system was identified by PADLOC as a phage defense candidate (PDC) M66 (Fig 5a) across numerous homologs of the tree. A similar domain architecture was also found recently in the experimentally validated system DS-6 [34]. In the RT tree, the association of the RT with HipA/DUF3037 occurred at least in six different cases (Fig 5b and 5c and S2 File) in two different clades (Fig 5b). One occurrence is found in species from the Bacteroidota phylum, and the other one is in species from the Pseudomonadota phylum (Fig 5b and 5c). Furthermore, we found that the C-terminal region of the DUF3037 shares sequence similarity to the DNA-binding protein from close Type III-A1 retrons (Fig 5d), likely representing a recent evolutionary event where the PRTase effector has been replaced by a predicted functional system (PDC-M66) formed by HipA and DUF3037. Although not very abundant, this case could represent a nascent association or even a retron that has captured new effectors, in an scenario similar to what occurred in Retron types II-A1 and I-A, where the NDT effector was replaced by a Septu defense system [21].

We found that the operon containing Retron-Sda1 exhibits defense against phage Bas51 when induced in bSLS114, which gave us the opportunity to test dependence on each component of the operon. Defense was unaffected by deletions of the HTH or ncRNA, or by introduction of mutations that inactivate the RT (dRT) (Fig 5e). However, deletion of the HipA kinase eliminated all defense. Furthermore, deletion of the DUF3037 gene caused cell toxicity. Combined, these results support a role for the HipA kinase and DUF3037 gene as a standalone defense system, where the HipA kinase functions as a toxin that is neutralized by the DUF3037 protein. This system does not require the retron RT or ncRNA for defense against Bas51.

We also explored the case of Retron-Bth1, which lacks an apparent effector gene and contains two RT genes. Unlike all other retrons isolated in this study which exist in the bacterial genome, we found Retron-Bth1 on an extrachromosomal assembly of 234kb that could potentially be a plasmid or phage. One RT gene in this operon belongs to a group II intron which appears to have inserted into the Retron-Bth1 RT gene. This retron was observed to produce RT-DNA in Fig 2b, so we hypothesized that the group II intron may splice out of the retron RT in frame, allowing for translation of functional protein to occur. To confirm that RT-DNA was produced by the retron RT alone, and not the group II intron RT, we cloned Retron-Bth1 in bSLS114 with the group II intron deleted (Fig 5f). This visibly increased RT-DNA production, demonstrating that the observed band derives from the retron RT. The clear difference in band intensity between the wild-type operon and the operon with the group II intron deleted demonstrates that group II intron splicing is inefficient in *E. coli* and can inhibit proper translation of the RT.

## Precise editing using recombitrons based on naturally sourced retrons

The recombitron approach that we used to edit Bas51 and identify its methyltransferase as the trigger was enabled by repurposing retron components into biotechnology for genome editing. In recombitrons, a modified retron ncRNA is reverse transcribed by the retron RT to produce RT-DNA that serves as a recombineering donor, where the RT-DNA is incorporated into the lagging strand during DNA replication, precisely changing the genome in the process [7–9,15,35]. This approach increases editing efficiencies compared to producing template DNA in vitro then delivering it to a cell and is particularly useful when editing phage genomes because the recombineering donor is produced in high abundance within the host [33]. We have found that this technology can be built on different retrons, but that some retrons support more efficient editing than others [22]. To determine whether naturally sourced retrons could be used to design efficient editors, we engineered them to produce templates for recombineering of the *lacZ* gene in *E. coli* (Fig 6a).

Naturally sourced retron RT-DNAs were modified to encode a template for a 10 bp deletion in the *lacZ* gene. For retrons without RT-DNA data, the RT-DNA was assumed to cover the longest stem-loop in the ncRNA secondary structure. Retrons were expressed on a plasmid also containing a gene for CspRecT, a ssDNA annealing protein that enables efficient recombineering from the reverse transcribed donor. Based on prior studies, donor sequences were inserted roughly 10

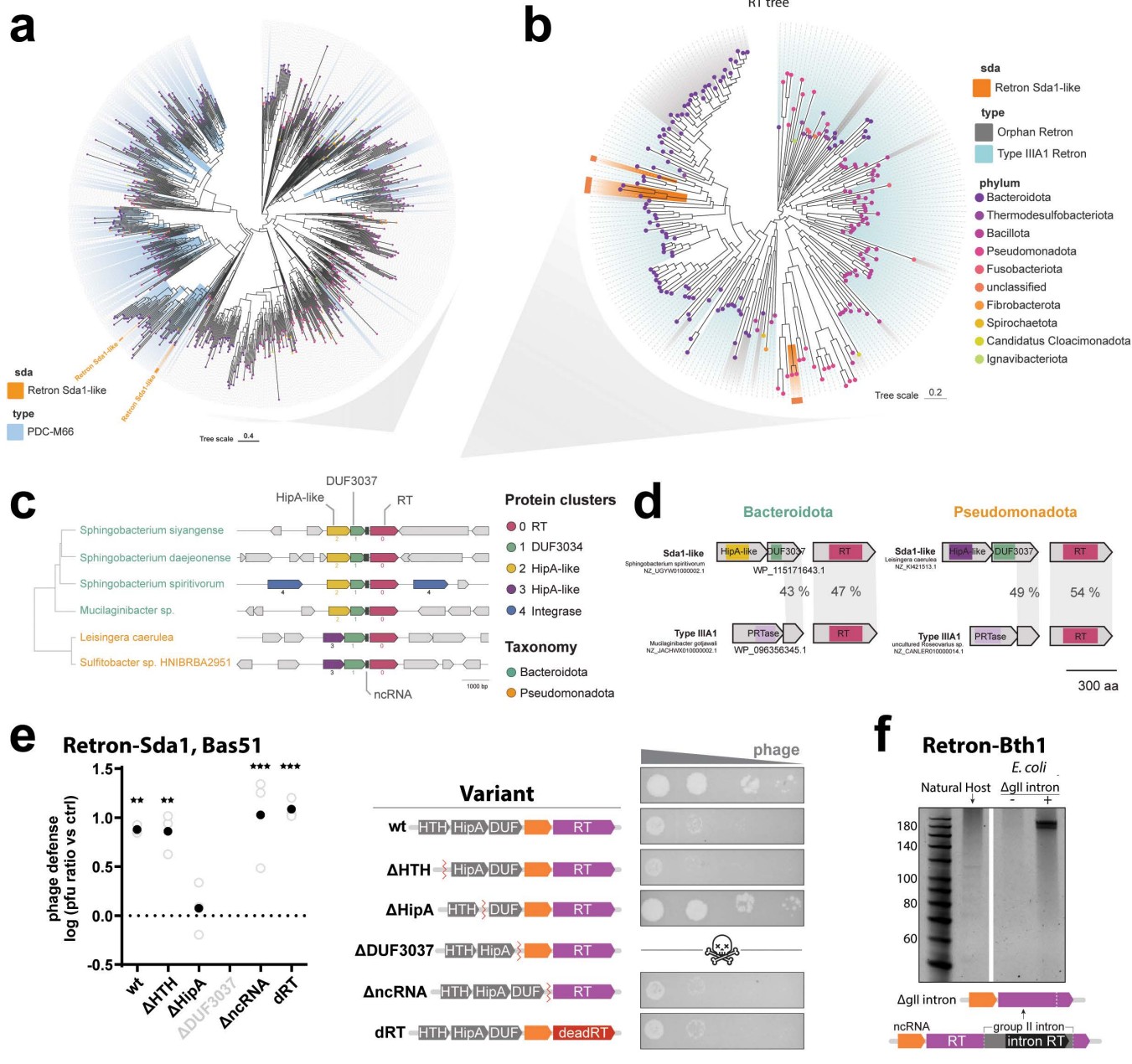

**Fig 5. Retron-Sda1 is an example of a previously unknown retron type. a)** Phylogenetic analysis of DUF3037-containing genomic neighborhoods. **b)** Phylogenetic analysis of RT homologs. **c)** Six examples of retron RT associated with DUF3034 and HipA-like proteins drawn from nonneighboring branches of the tree in **b. d)** Sequence homology between the c-terminus of the DUF3037 gene and the PRTase effector normally associated with Type III-AI retrons. **e)** Phage defense (log transformed) calculated as the pfu (plaque forming units) per ml on a control strain not expressing the retron divided by pfu per ml on the retron expressing strain, for Retron-Sda1 against Bas51. For each variant, open circles show each of three biological replicates and the closed circle shows the mean. One-way ANOVA $P = 0.002$, Dunnett's follow-up testing (corrected for multiple comparisons) empty plasmid vs. wt ($P = 0.0031$), ΔHTH ($P = 0.0037$), ΔncRNA ($P = 0.0009$), dRT ($P = 0.0006$), all other conditions ns. Variants are explained by schematic in the middle with example plaque assays to the right for each variant. Deletion of DUF3037 created a toxic construct where phage defense could not be quantified. Additional statistical details in S3 Table. The data underlying this panel can be found in S4 Table. **f)** RT-DNA production from PAGE analyses of Retron-Bth1 in its native host and in *E. coli* with the gII intron deleted, respectively (uncropped gel in S1 Raw Images).

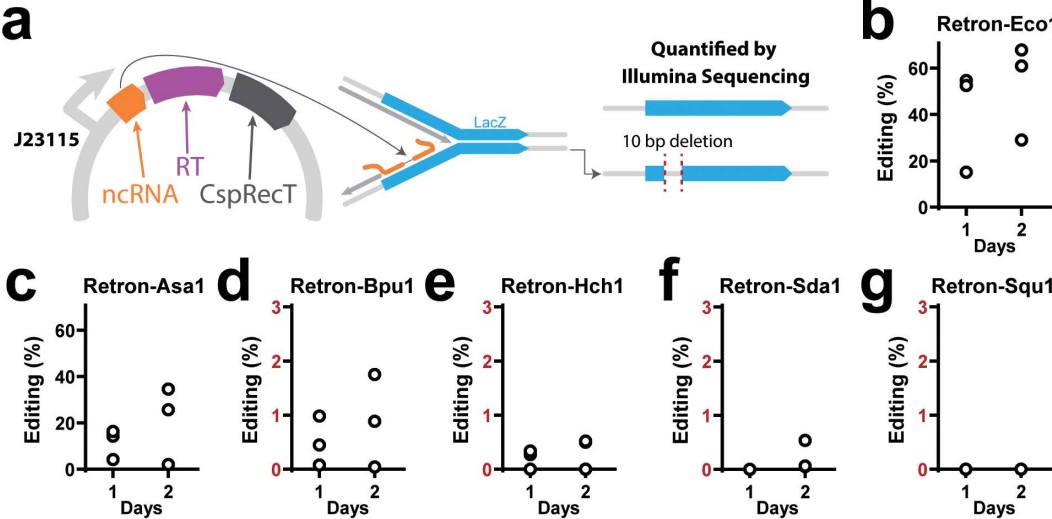

**Fig 6. Precise editing using recombitrons based on naturally sourced retrons. a)** Left, schematic of plasmid for recombineering in *E. coli*; right, schematic of recombineering LacZ gene with 10 bp deletion. **b)** Percent of cells with LacZ gene precisely edited using the Eco1 recombitron after culturing for 1 or 2 days, quantified using Illumina sequencing. **c)** Experiments and plots identical to **b**, but using the Asa1 **(c)**, Bpu1 **(d)**, Hch1 **(e)**, Sda1 **(f)**, and Squ1 **(g)** recombitrons. Additional statistical details in S3 Table. All data underlying this figure can be found in S4 Table.

base pairs out from the base of the main RT-DNA stem loop, replacing the remainder of the stem and the loop. Editing data was quantified by Illumina sequencing.

The Retron-Eco1 editor, which uses a well-studied retron from *E. coli* as a scaffold, exhibited higher editing rates than editors using naturally sourced retrons (Fig 6b). Among editors based on naturally sourced retrons, the Retron-Asa1 editor was the most efficient and made deletions in ~30% of cells by the second culture round (Fig 6c). The remaining four retron editors exhibited much lower editing rates (below 2% after one day) (Fig 6d–6g). Interestingly, Retron-Asa1, which displayed the highest editing rate, also consistently produced more abundant RT-DNA when compared to retrons with lower editing rates (S3 Fig).

## Discussion

Though many retrons have been studied in *E. coli*, few have been identified from environmental samples or studied in their native bacterial hosts. This study describes a new approach to isolate retrons and their native hosts from environmental soil and water samples by screening for RT-DNA bands on polyacrylamide gels. This approach expands the collection of known retron hosts and could enable studies of retrons that cannot be transformed into *E. coli* or who function differently outside of their natural host bacterium.

For purely practical reasons, we chose to grow bacteria mainly at 37 °C in LB media in this study. These culture conditions undoubtedly select for faster-growing organisms and differ from the natural environments from where we sourced these bacteria. Indeed, the overrepresentation of type II-A and III-A retrons may reflect selection due to culturing conditions favoring bacteria harboring these types. Future work that tests a wider variety of culture conditions, including those that mimic natural environments, may find additional interesting retron-bearing bacteria and we hope that this work provides a roadmap for isolation of many new retron hosts from the wild. We additionally used the natural codons for all testing, including for expression in *E. coli*. We found RT-DNA production for six of the seven systems, with the last system being toxic to cells, indicating that proteins are being produced in all cases. However, it is possible that optimizing codon usage could have increased the potency of some systems, which we did not test.

PADLOC identified retron operons in whole genome sequencing data of most samples where clear, putative RT-DNA bands were visible on polyacrylamide gels. However, the case of Retron-Bth1 illustrates a more unique example where the physical identification of RT-DNA led us to describe a piece of biology that would have likely been overlooked in metagenomics. First, it is found on an extrachromosomal element and the fact that we had the strain let us link this element to the species. Second, because the RT gene is disrupted by the insertion of a group II intron and is not associated with a canonical effector, it is not readily identified as a retron by annotation tools. We would likely have not identified this odd, disrupted, but still DNA-producing retron, if we had not found the RT-DNA in its natural host and gone searching deeply for RTs in the genome.

Interestingly, Retron-Sda1 contains a type III retron RT gene but without its typical PRTase (phosphoribosyltransferase-like) effector. Instead, this effector appears to have been replaced with a different phage defense system composed of a HipA-like kinase domain and DUF3037 domain that function on their own [34]. Although the RT-DNA and RT gene are not involved in defense against the phage we tested, this new association could represent an early stage in the evolution of a new retron type, given the wide-ranging association of retron systems with different effector types [21].

We also show that the N6 adenine methyltransferase in Bas51 acts as a trigger for Retron-Squ1, while both the N6 adenine methyltransferase and glycosidase contribute to triggering Retron-Asa1. This suggests that these naturally sourced retrons are triggered by methylation of the RT-DNA, as has been shown for other retrons [1], whereas the glycosidase perturbs the retron effector-bound complex via an unknown mechanism. We provide additional evidence for triggering by the methyltransferase gene through identification of methylation site motifs on the ncRNA of both retrons, where it is found specifically in the reverse transcribed region for Retron-Asa1. For Retron-Squ1, we confirm that methyltransferase activity also triggers the retron in its native host, *S. quinivorans*, and that the degree of retron-induced toxicity increases with retron copy number.

While most naturally sourced retrons isolated in this study are not efficient editors in *E. coli*, the Asa1 recombitron exhibits gene editing potential. Further research could explore the gene editing capabilities of recombitrons in other cell types or conditions more similar to their native host bacteria and environment.

## Methods

### Isolating bacteria from soil and water samples

Small samples of soil (~2.5 g) and water (~10 mL) were collected from glacial lakes and streams as well as alpine mountains, woodlands, and parks. Microbes were isolated from soil samples using the Zymo fecal/soil microbe kit, which removed dirt and rocks. Water and soil processed with the Zymo kit were then plated on plain LB and allowed to grow overnight or until colonies formed at 37 °C.

### Identifying RT-DNA bands on polyacrylamide gel

Colonies were picked, sampling from those with visibly different appearances, and cultured in 3 or 5 mL of LB overnight, shaking at 37 °C. Cultures were then passaged with a 1:100 dilution into 3 or 5 mL of LB and cultured for another 7 h, shaking at 37 °C. DNA from these cultures was miniprepped using the QIAprep Spin Miniprep kit (Qiagen 27104) and analyzed on a 15% TBE-urea (Thermo Fisher EC6885) gel to visualize possible RT-DNA bands. Based on the length of previously identified retron RT-DNA [22], bands between 40 and 200 base pairs were considered as a potential signature of a retron.

### Isolating RT-DNA and sequencing

In samples with potential RT-DNA bands, DNA was isolated from cultures using the Qiagen kit and ssDNA was isolated from the miniprep using the ssDNA/RNA clean and concentrator kit (Zymo Research). RT-DNA was prepped for sequencing by taking the resulting material and extending the 3′ end in parallel with dATP, using terminal deoxynucleotidyl

transferase (TdT) (NEB). This reaction was carried out in 1 × TdT buffer, with 60 units of TdT and 125 M dATP for 60 s at room temperature with the aim of adding ~25 adenosines before inactivating the TdT at 70 °C for 5 min. Next, a polynucleotide anchored primer was used to create a complementary strand to the TdT extended products using 15 units of Klenow Fragment (3′→5′ exo-) (NEB) in 1 × NEB2, 1 mM dNTP and 50 nM of primer containing an Illumina adapter sequence, nine thymines (for the pA extended version) or six guanines (for the pC extended version), and a non-thymine (V) or non-guanine (H) anchor. The product was cleaned using a Qiagen PCR cleanup kit and eluted in 10 µl water. Finally, Illumina adapters were ligated on at the 3′ end of the complementary strand using 1 × TA Ligase Master Mix (NEB). All products were indexed and sequenced on an Illumina MiSeq instrument.

## Whole genome sequencing and identification of retron operons

Genomic DNA was isolated from overnight cultures of host bacteria using the Zymo Quick DNA/RNA Miniprep Kit (Zymo D7001). Genomic DNA was then tagmented with Tn5 transposase preloaded with Illumina sequencing adapters, indexed, and sequenced on a NextSeq 2000. Whole-genome contigs for *Pseudomonas peli*, *Aeromonas salmonicida*, and *Sphingobacter daejeonense* samples were assembled using Geneious Prime software. Sequencing data for the remaining samples included more than 5.5 million reads, so we used a Velvet plugin instead to assemble whole-genome contigs more quickly. In the *H. chinensis* and *S. quinivorans* samples, sequencing yielded 40 million and 24 million reads, respectively, but only 10% and 50%, respectively, were used for Velvet assemblies.

RT-DNA reads were mapped to whole-genome data to locate the potential retron operon. Genomes were additionally analyzed by PADLOC [28], which identifies defense systems, including retrons, based on known motifs. PADLOC also identifies the likely start and end of accessory and RT genes.

In samples where whole-genome data was not collected or where assembled contigs did not contain the entire retron operon, genomic DNA was tagmented using Tn5 preloaded with Illumina sequencing adapters and amplified to enrich for retron-containing fragments. To perform this PCR, two primers were designed to bind the known RT-DNA sequence facing opposite directions to capture both ends of the retron. Another two primers were designed to bind the two Illumina adapters at each end of the fragment. Four PCR reactions were then performed, one for every combination of RT-DNA-binding primer and adapter-binding primer. The enriched sample was then sequenced using either MiSeq or Sanger sequencing of the resulting amplicons. When necessary, this process was repeated to capture the entire retron operon with primers moving successively outward from the known part of the retron.

## Identifying new retron types in sequencing data

In samples with potential RT-DNA, but where PADLOC did not identify full retron systems, potential retron RT genes were identified using the PADLOC, DefenseFinder, and NCBI's BLAST programs to find sequences with similarity to known retron RTs. Possible retron accessory genes were identified by studying both their direction and gene functions predicted by BLAST and EMBL-EBI's Interpro tool.

## Finding homologs of Retron-Sda1

To investigate the unusual genomic context of Retron-Sda1, we conducted a comprehensive homology and phylogenetic analysis. Initially, protein sequences corresponding to the RT and DUF3037 domains of Retron-Sda1 were extracted from the assembled genome contigs. These sequences were used as queries in a BLASTp search against the NCBI non-redundant (nr) protein database [https://ftp.ncbi.nlm.nih.gov/blast/db/FASTA/nr.gz] to identify homologous proteins across diverse bacterial taxa. Multiple sequence alignments of the retrieved RT and DUF3037 protein sequences were performed using MAFFT v7.475 with the G-INS-i algorithm [36]. Phylogenetic trees were constructed using FastTree v2.1 [37] with default parameters. To examine the genomic context of the identified homologs, we utilized the NCBI Entrez Programming Utilities [38] API to retrieve genomic neighborhoods surrounding each homologous RT and DUF3037 gene. Genomic

regions extending ±10 kilobases from each gene were extracted and analyzed for co-localized genes encoding HipA-like kinases and DUF3037 domain-containing proteins. Defense system annotation was carried out using PADLOC v2.0.0 [27], This analysis was facilitated by custom Python scripts utilizing the Biopython library [39]. Genomic neighorhoods were visualized using a custom JavaScript code and made availale as a portable HTML file.

## Cloning retron plasmids and verifying RT-DNA production in *E. coli*

Predicted retron operons were amplified by PCR from their host bacterial genomes and cloned into a pET-21(+) backbone (ColE1 ori, kanamycin resistance) in NEB 5-alpha Competent *E. coli*. Plasmids were verified with Sanger sequencing and then transformed into BL21$^{\Delta Eco1}$ (bSLS.114 [7], Addgene #191530).

Colonies were picked into 3 mL of LB, grown overnight (shaking at 37 °C), passaged at a 1:100 dilution into 3 mL of LB, and then induced with working concentrations of 1 mM IPTG (GoldBio) and 2 mg/ml L-arabinose (GoldBio). After 7 h of culture with induction, DNA was miniprepped using the QIAprep Spin Miniprep kit and RT-DNA bands were visualized using electrophoresis on 15% TBE-urea gels (Thermo Fisher EC6885).

## Phage strains and propagation

Lambda and T5 phages were initially propagated from ATCC stocks (Lambda WT #23724-B2, T5 #11303-B5). T4 phage was drawn from a previously established stock [33]. MG1,655 *E. coli* culture was infected with phage and grown overnight. The culture was then centrifuged for 10 min at 3,434$g$ and the supernatant was filtered at 0.2 μm to remove bacterial remnants. Lysate titer was determined using the full plate plaque assay method as described by Kropinski and colleagues [40]. We used a strictly lytic version of lambda carrying two early stop codons in the cI gene, responsible for lysogeny control, to ensure the phage was strictly lytic (lambda ΔcI) [33]. This lambda strain additionally contains a genomic deletion between positions 21,738 and 27,723 that includes genes involved in lysogeny control. Lambda ΔcI was used for all phage defense experiments. Bas51 phage was propagated from the Leibniz Institute DSMZ stock (https://www.dsmz.de/collection/catalogue/details/culture/DSM-112929), and was originally isolated in the BASEL phage collection [31].

## Testing retron defense against *E. coli* phages

MG1,655 cells were transformed with plasmids containing the retron operons and plated. Transformed colonies were picked and cultured overnight in 3 mL of LB (shaking at 37 °C) then passaged in 3 mL of MMB (LB medium supplemented with 0.1 mM MnCl$_2$ and 5 mM MgCl$_2$) for 3−5 h. 200 μL of culture was plated in 3 mL of MMB soft agar in each well of a 4-well MMB agar plate. Lambda, T4, T5, and Bas51 phages were serially diluted to $10^{-4}$, $10^{-5}$, $10^{-6}$, and $10^{-7}$ in MMB media. 5 μL of each dilution was spotted on these plates. After the spots have completely dried, plates were incubated at 37 °C overnight. Resultant plaques were counted to determine the titer (pfu/mL) of each phage strain on each retron-expressing bacterial lawn.

## Isolating mutant phages and identifying retron trigger genes

For retron and phage combinations where defense occurred, phages and retrons were plated on MMB soft agar at the lowest phage dilution where few or no plaques were visible. Plaques were then picked from plates and, along with the wild-type phage, propagated in a culture of the MG1655 transformed with the given retron. These cultures were plated in MMB soft agar and grown overnight. SM buffer (100 mM NaCl, 8 mM MgSO$_4$, 50 mM Tris-HCl pH 7.5) was added to submerge the wells and left for 4 h at 4 °C to extract phages. The buffer was then removed and filtered with a 0.2 μm filter. For amplification-free sequencing, extracellular DNA was removed through DNase I treatment, with 20 U of DNase I (NEB, M0303S) per 1 mL of phage lysate, incubated at room temperature for 15 min and then inactivated at 75 °C for 5 min. Phage were then lysed and DNA extracted using the Norgen Phage DNA Isolation Kit (Norgen, 46800). Phage mutant samples were prepped for nanopore sequencing using the Ligation Sequencing Kit (SQK-LSK109) and Native

Barcoding Expansion (EXP-NBD196) from Oxford Nanopore Technologies (ONT). Nanopore sequencing was performed using an R9.4.1 flow cell (FLO-MIN106D) on a MinION device. Wild-type phage samples were prepped using the Ligation Sequencing Kit (SQK-LSK114) and sequenced using an R10.4.1 flow cell (FLO-PRO114M) on a PromethION device. Base calling was performed using ONT's Guppy (for phage mutants) or Dorado (for wild type phage) Basecaller software on the high accuracy setting.

Nanopore sequencing reads for mutant and wild-type phage were assembled independently using the Geneious Prime software and consensus sequences were mapped to reference genomes to identify mutations. Genes containing mutations in multiple phage escapees for a given retron were considered potential trigger genes. To confirm potential trigger genes, wild-type phage was engineered to incorporate the individual mutations seen in the escapee (see Phage recombineering below). These engineered phage mutants were then tested for defense in the same retron-expressing bacterial strain they were isolated from (see Testing retron defense against E. coli phages above).

## Phage recombineering

To make each phage mutant, a recombitron plasmid (as described in Fishman and colleagues [33]) with a 90 nt editing donor was co-transformed with pORTMAGE (contains CspRecT and mutL E32K accessory genes for recombineering) into a modified strain of *E. coli* MG1655 with inactivated *exoI* and *recJ* genes, as well as an arabinose-inducible T7 polymerase (bMS.346 [7], Addgene #220588). Individual transformants were picked from plates and grown in 3 mL LB media overnight in a shaking incubator at 37 °C. The following day, these bacterial editor cultures were diluted 1:100 in 3 mL MMB media containing 1 mM IPTG (GoldBio), 2 mg/ml L-arabinose (GoldBio), and 1 mM m-toluic acid (Sigma-Aldrich) to express the recombineering machinery. Passaged cultures were grown for an additional 2–3 hs at 37 °C, then diluted to an OD600 of 0.2 in a fresh culture of 3 mL MMB with inducers. Wild-type phage lysate was spiked into these cultures at a MOI of 0.1. The infected cultures were then grown for 16–18 h in a shaking incubator at 37 °C. Cultures were then centrifuged for 10 min at 3,434*g* and the supernatant was filtered at a 0.2 μm pore size to obtain clarified phage lysate. To increase the percent of edited phage in the population, this recombineering protocol was repeated using the newly isolated phage lysate as the spike-in for a fresh batch of bacterial editors. Editing rates were monitored after every round of recombineering by PCR amplifying the phage lysate with primers flanking the edit site and then Sanger sequencing those amplicons.

After two rounds of recombineering, individual phage mutants were isolated by plaquing on a lawn of bacterial editors. To do this, editors were grown overnight and passaged in 3 mL MMB with inducers for 3–5 h. Edited phage lysate was diluted 1:10,000 in MMB media. Then, 5 μL of diluted lysate was spiked into 1 mL of passaged editor culture, all of which was mixed into 20 mL of soft agar and poured onto a LB plate. Soft agar plates were incubated at 37 °C overnight. The next morning, eight individual plaques were picked for each phage mutant, resuspended in 100 μL of SM Buffer, and left at 4 °C for at least 4 h to allow the phage to disseminate into the buffer. Amplification of the edit site followed by Sanger sequencing was again performed to screen for lysates containing the desired edit. Those lysates were further propagated to obtain high-titer phage mutants for use in subsequent experiments.

## Identification of methylation sites in phage genomes

To identify N6-methyladenines in the Bas51 phage genome, we extracted phage genomic DNA (see Isolating Mutant Phages and Identifying Retron Trigger Genes) and sequenced on a PromethION device using the Ligation Sequencing Kit (SQK-LSK114) and R10.4.1 flow cell (FLO-PRO114M) from Oxford Nanopore Technologies (ONT). Base calling with performed using ONT's Dorado software on super accuracy (SUP) settings with 6mA modification calling enabled. Reads were then aligned to a Bas51 reference genome using Dorado. Methylated adenines were extracted using Modkit (ONT), filtering for sites with>100x coverage and >60% methylation frequency. To generate logo plots, the flanking sequences 10 nucleotides upstream and downstream of all sites passing the filter was extracted and fed to WebLogo3 software [41]. Modkit's motif search tool was used to quantify methylated adenines found in GATC and GTAC motifs.

## Methyltransferase trigger assays

A plasmid containing the Bas51 N6 adenine methyltransferase was constructed by amplifying the gene from phage lysate and cloning it into a CloDF13 backbone under the Lac promoter using Gibson assembly. These were transformed into NEB 5-alpha Competent *E. coli* and clones were verified using Sanger sequencing.

A ΔSqu1 strain of *S. quinivorans* was built using lambda Red-mediated recombineering [42]. Briefly, a FRT-flanked chloramphenicol-resistance gene was amplified with primers that add 50 bp homology arms to the regions immediately flanking the Squ1 operon in *S. quinivorans*. The amplicon was gel-purified and electroporated into a strain of *S. quinivorans* expressing the lambda Red system. Correct clones were selected for by plating on LB agar containing chloramphenicol and verified via Sanger sequencing of the Squ1 locus.

To perform the trigger assay, plasmids containing the methyltransferase (or an empty vector control) and/or Retron-Squ1 (or an empty vector control) were electroporated into *S. quinivorans* using the same electroporation settings and protocol as for *E. coli*, except that all incubations were done at 30 °C. Transformants were plated on LB agar containing the appropriate antibiotics and 1% glucose for repression of the Lac promoter. Three colonies were picked for each sample (biological replicates) and grown overnight in 0.5 mL LB media + antibiotics + 1% glucose in a deep 96-well plate (Thermo Scientific 260251) covered with an air-permeable seal (Breathe-Easier, Diversified Biotech) at 30 °C with shaking. The next morning, cultures were diluted 1:100 in 200 µL of LB media + antibiotics in a black, clear-bottom 96-well microtiter plate (Corning 3603). The plate was covered with a transparent air-permeable seal (Breathe-Easy, Diversified Biotech BEM-1) and incubated in a microplate reader (Tecan Infinite 200 PRO) with shaking at 37 °C for 18 hrs with $OD_{600}$ measured every 10 min. At the 2 h time point, IPTG was added to all samples to induce the Lac promoter.

## Engineering retrons for recombineering in *E. coli*

To edit bacterial genomes using recombineering, retrons were modified to encode a recombineering donor in the reverse transcribed region of the ncRNA. This 70 nt donor incorporates a 10 nt deletion in the *lacZ* gene. Effector proteins were excluded from retrons for recombineering. The modified retrons were cloned into the pACYC-Duet1 backbone (p15A ori, chloramphenicol resistance) where they are co-expressed with CspRecT under the constitutive J23115 promoter. These plasmids were then transformed into bMS.346 cells and immediately inoculated into 5 mL LB cultures, which were grown overnight, shaking at 37 °C. The next day, 25 µl of culture was collected, mixed with 25 µl of water, and boiled at 95 °C for 10 min. The same cultures were then passaged at a 1:100 dilution in fresh 5 mL LB and grown overnight again, after which another set of boiled lysates was collected. 0.5 µl of each boiled lysate from Day 1 and Day 2 of editing was then used as a template in 25 µl PCR reactions with primers flanking the edit site, which additionally contained adapters for Illumina sequencing preparation. These amplicons were indexed and sequenced on an Illumina MiSeq or NextSeq 2000 instrument. Sequencing reads were processed with CRISPResso2 [43] (https://github.com/pinellolab/CRISPResso2) to quantify the percentage of precisely edited genomes. The following equation was used to calculate percent editing:

$$\% \ edited = 100 * \frac{\# \ of \ reads \ aligning \ to \ edited \ sequence}{\# \ of \ reads \ aligning \ to \ edited \ sequence + \# \ of \ reads \ aligning \ to \ wild \ type \ sequence}$$

## Supporting information

**S1 Fig. Phage defense testing of retrons-Hch1 (a) and -Bpu1 (b) in bSLS.114 with overexpression.** Images are matched phage plaque assays for each phage with C indicating the control condition and R indicating the retron-expressing condition.
(TIF)

**S2 Fig. Logo showing the predominant motif of methylated adenine sites in Bas51 phage with a T83P mutation in the N6 adenine methyltransferase.**
(TIF)

**S3 Fig. Replicates of PAGE analysis of RT-DNA production in *Escherichia coli*, induced and uninduced.**
(TIF)

**S1 File. Interactive HTML file showing the genomic neighborhoods of all DUF3037 homologs.**
(HTML)

**S2 File. Interactive HTML file showing the genomic neighborhoods of the six Sda1-like retrons.**
(HTML)

**S1 Table. Sample characteristics and retrons identified.**
(XLSX)

**S2 Table. Additional statistical details.**
(XLSX)

**S3 Table. Retron operons.**
(XLSX)

**S4 Table. Data.**
(XLSX)

**S1 Raw Images. Uncropped gels.**
(PDF)

## Acknowledgments

We thank Sofia Arreola, Kate Crawford, and Chloe Fishman for piloting methods for ssDNA extraction from environmental bacteria prior to this study.

## Author contributions

**Conceptualization:** Seth L. Shipman.

**Formal analysis:** Mario R. Mestre, Seth L. Shipman.

**Funding acquisition:** Seth L. Shipman.

**Investigation:** Kazuo L. Nakamura, Karen Zhang.

**Methodology:** Kazuo L. Nakamura, Karen Zhang, Matias Rojas-Montero, Seth L. Shipman.

**Project administration:** Seth L. Shipman.

**Software:** Mario R. Mestre.

**Supervision:** Karen Zhang, Matias Rojas-Montero, Seth L. Shipman.

**Validation:** Kazuo L. Nakamura, Karen Zhang.

**Visualization:** Seth L. Shipman.

**Writing – original draft:** Kazuo L. Nakamura, Karen Zhang, Mario R. Mestre.

**Writing – review & editing:** Seth L. Shipman.

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
