## [Editor Report · Decision Letter 0]

28 Jan 2025

Dear Seth,

Thank you for submitting your manuscript entitled "Phage defense and genome editing using novel retrons sourced from isolated environmental bacteria" for consideration as a Research Article by PLOS Biology.

Your manuscript has now been evaluated by the PLOS Biology editorial staff as well as by an academic editor with relevant expertise and I am writing to let you know that we would like to send your submission out for external peer review.

Once your full submission is complete, your paper will undergo a series of checks in preparation for peer review. After your manuscript has passed the checks it will be sent out for review. To provide the metadata for your submission, please Login to Editorial Manager (https://www.editorialmanager.com/pbiology) within two working days, i.e. by Jan 30 2025 11:59PM.

Best wishes,

Melissa

Melissa Vazquez Hernandez, Ph.D.

Associate Editor

PLOS Biology

---

## [Decision Letter · Decision Letter 1]

12 Mar 2025

Dear Seth,

Thank you for your patience while your manuscript "Phage defense and genome editing using novel retrons sourced from isolated environmental bacteria" was peer-reviewed at PLOS Biology. Your manuscript has been evaluated by the PLOS Biology editors, an Academic Editor with relevant expertise, and by three independent reviewers.

As you will see in the reviewer reports, although the reviewers acknowledge the potential interest in your findings, they have also raised a substantial number of crucial concerns. Based on their specific comments and following discussion with the Academic Editor, it is clear that a substantial amount of work would be required to meet the criteria for publication in PLOS Biology. Given our and the reviewer interest in your study, we would be open to inviting a comprehensive revision of the work. Reviewer 1 has some suggestions to improve the text and figures. Reviewer 2 appreciates the work but says that the study is preliminary and lacks explanations as well as validation of the newly found retrons. The reviewer particularly mentions one retron that was found on the biological samples but not on PADLOC, yet it was not explored. Same thing happened with other retrons that did not do any wet lab validation. Another point mentioned is the lack of further experiments regarding the activity of the methyltransferase and the glycosidase. Reviewer 3 is also positive about the relevance, but also thinks further experiments are necessary. This reviewer specifically suggests to test the capacity of genome editing of the retrons in their native hosts, or at least in a broader range of bacterial species and would like you to test the retrons against more phages, like the Basel Phage Collection, and again test in their native hosts.

IMPORTANT: After discussion with the Academic Editor and the cross-comments, you should aim to address all of the textual/formatting changes recommended by the reviewers. In addition, we will expect further experimental validation of their retron systems, as suggested by reviewers 2 and 3. However, we do not expect you to complete the genome editing in a non-E.coli background as this could be highly challenging, and might be outside of the scope of the study.

Given the extent of revision that would be needed, we cannot make a decision about publication until we have seen the revised manuscript and your response to the reviewers' comments. Your revised manuscript would need to be seen by the reviewers again, but please note that we would not engage them unless their main concerns have been addressed.

We appreciate that these requests represent a great deal of extra work, and we are willing to relax our standard revision time to allow you 6 months to revise your study. Please email us (plosbiology@plos.org) if you have any questions or concerns, or envision needing a (short) extension.

**IMPORTANT - SUBMITTING YOUR REVISION**

*Resubmission Checklist*

*Published Peer Review*

*PLOS Data Policy*

*Blot and Gel Data Policy*

Sincerely,

Melissa

Melissa Vazquez Hernandez, Ph.D.

Associate Editor

PLOS Biology

REVIEWERS' COMMENTS:

Reviewer #1:

It was a pleasure to review this creative study by Nakamura and colleagues on the characterization of retron elements from environmental bacteria. Their unbiased sampling approach impressively demonstrates the abundance of these systems, and the in vivo identification is a rare and valuable contribution to the phage defense field, which has largely focused on in silico identification. This is particularly important as it allows for the study of defense systems in their native host backgrounds, a significant limitation of most current research. The manuscript is well written, and the data and figures are of high quality. I have the following suggestions for improvement:"

Even though the gene color scheme is labelled in 1d, i.e. pink arrow = effector, it would be helpful to clearly state this scheme at end of the legend. It took me two passes to understand the significance of the colors. Additionally, the abbreviations used in the functional annotations should be expanded in the legend, e.g. WH, PRTase, RT etc. It is unclear to me which of these are the DNA-binding domain containing proteins mentioned in the following paragraph.

Again in Figure 1E, why is the DUF3037 gene colored as an effector? Do the authors hypothesize this protein acts as the effector in this system and what is the evidence?

Text beginning at paragraph "Retron-Sda1 is an example…" goes immediately into retron classifications. For the non-expert reader, it would be helpful to include some text in the introduction to contextualize the classification of retron systems i.e. how many are there are what are the classification criteria? There is some text related to this in the fourth paragraph of the discussion, but it came too late to help me contextualize the data in Figure 1 and associated paragraph.

More information should be given on how PDC-M66 was classified as a putative defense system classification (in the second paragraph following Figure 1). What criteria was used to make this prediction? I could not find any publications mentioning PDC-M66.

In the third paragraph after Figure 3 "However, the Asa1 retron showed a subtle phenotype…". Should this say Squ1 as it appears from the figure?

As Bas51 is not a well known coliphage, it would be helpful to give some more information about this phage to contextualize its sensitivity to Asa1 and Squ1. Is it related to any of the other phages used? I.e. does it share any similarity with phage T5?

I found the second paragraph of the section titled "Defense Mechanisms of Retron-Asa1 and -Squ1" hard to follow. I infer that 10 escape phages were sequenced in total but only 6 were found to be unique after polynucleotide and synonymous changes were ignored. If this is correct, please add a sentence stating this clearly. A symbol next to the tracks in figure 5 a&b could be added to indicate escapees found to have redundant mutations.

No data is cited for the statement "Retron-Asa1 recombitron's editing rates are likely partly a result of its RT-DNA production in E. coli…". Does this reference figure 3B? Can the abundance of RT-DNA by Asa1 be reproduced in biological replicates?

There is a citation missing for the statement "the majority do and should be identifiable when screening for RT-DNA bands."

I found the following typographical errors:

"retrons have also be repurposed"

Figure three legend "E. Coli" (change to lowercase c)

Reviewer #2:

The study by Nakamura, Zhang, et al. isolates new retrons from natural bacterial isolates and includes a basic characterization of some of these in bacterial immunity and genome editing. I have great sympathy for this work, but am unhappy about its currently very streamlined style that lacks a variety of content and explanations which would make it more convincing and accessible to non-experts. Most importantly, the two most interesting newly found retrons are only briefly mentioned but not at all explored or validated in wet lab experiments. This is very confusing.

The introduction is comparably short and completely focused on retrons as phage defense elements (and their research history) without any general context. My view is that this introduction may be appropriate for a specialist audience but should be widened to make the manuscript accessible also to readers from outside the field of bacterial immunity. I also missed a convincing argument for the relevance of this work beyond phage defense aficionados. The manuscript essentially says (third paragraph) that only retrons from a couple of hosts have been studied so far, but it fails to highlight why readers from outside the field should care about the characterization of additional ones. Of course there are various good reasons for this (e.g., the acquisition of better tools for molecular genetics) but I feel that this should be spelled out.

The authors began their study with the isolation of retron-carrying hosts from natural samples based on the cultivation of environmental microbes and the subsequent screening for characteristic ssDNA bands on polyacrylamide gels. I do not doubt that this approach gave the authors some new retrons. However, the methodology of how they got there seems rather crude and very arbitrary in the sense that the procedure and possible limitations are not explained or justified in any way (also not in the discussion section). As an example, the authors used rich LB medium at 37°C for the isolation of bacteria from mostly nutrient-poor, ambient-temperature environments. This does not seem very intuitive to me and could greatly bias the selection towards very few highly resilient, fast-growing organisms out of the vast diversity of environmental microbes.

I really liked the idea to screen for retron-expressing microbes using polyacrylamide gels because this opens the door to finding completely new, exotic kinds of retrons. However, the authors later largely relied on the standard tool PADLOC to identify the retrons in their isolates' genomes. While they could occasionally refine the PADLOC annotation using their experimental data, this makes the reader wonder why one should at all isolate environmental bacteria one by one to find new retrons. This would be much easier by simply putting hundreds of new metagenomic microbial genomes into PADLOC to get their retron loci (which could then be synthesized and studied). None of these important considerations are made anywhere in the study. Furthermore, the potentially most interesting discovery of this study - a non-identified organism (apparently isolated twice) which shows a retron-like band on gel but for which PADLOC found no retron - is only briefly mentioned but not discussed or explored at all. This organism may encode a completely new type of retron, and I think that this should be explored further. If there are clear RT-DNA bands visible on polyacrylamide gels (as highlighted by the authors), it would be easily possible to isolate and sequence these DNA molecules in order to map them back onto the genome.

From their natural samples, the authors found six different retrons in cultivated organisms which they studied further following standard procedures in the field. Initially, they focused on one specific retron (Retron-Sda1) which they analyzed bioinformatically because it seems to encode a different potential effector of abortive infection compared to what would be common for this type of retrons (type III). This is in principle interesting. However, the authors then specifically exclude this exact retron from any wet lab experiments testing ssDNA production in E. coli or bacterial immunity. This is not explained in any way beyond the authors' puzzling statement that "We also did not further study the III-A1-like Retron-Sda1".

After the bioinformatic analysis of Retron-Sda1 (without wet lab validation), the authors cloned their other new retron variants to show biological activity using ectopic expression in the model organism E. coli. Possible limitations of this host for the diverse retrons from very different other bacteria are neither explored nor mentioned anywhere. Not unsurprisingly, enforced expression of these retrons using T7 RNA polymerase in BL21-AI results in detectable retron DNA production (Figure 3). The gel also shows the well-characterized Eco1 retron - possibly as a control - which is neither mentioned nor explained in this context (but pops up again later in a different context with some more explanations).

Subsequently, the authors used the same plasmids to test a possible activity of their new retrons against four different E. coli phages. I was very confused by these experiments because their model organism - E. coli K-12 MG1655 - does not encode the T7 RNA polymerase that they used in the BL21-AI strain to overexpress the retron loci. Did the authors now rely on the natural promoters of these loci, and if yes, is there any evidence that these may be functional / expressed in E. coli? From their data, the authors claim that one of the retrons, Asa1, causes smaller plaques of phage T5 (which would be clear evidence of bacterial immunity) but do not provide a quantification of plaque size - this needs to be added (possibly also for Squ1 where at least on the image the effect looks much stronger). A suggested "strong trend of defense" of Asa1 against phage Bas51 seems to be largely due to two replicate experiments (Figure 4G), though the effect on plaque formation (Figure 4H) is quite clear. Evidently, the Squ1 retron defends against phage Bas51 by strongly reducing plaque counts.

Following standard procedures in the field, the authors then isolate immunity-resistant phage mutants to identify phage genes in Bas51 which may encode triggers of Asa1 and Squ1 retron immunity or interact with retron immunity in another way. For both retrons, the authors isolated phages with mutations in both an N6 adenine methyltransferase and a glycosidase gene. It is implied (but not shown or discussed) that these mutations may cause full or partial loss of function. Using targeted mutations in the phage genome, the authors convincingly argue that the methyltransferase (likely by modifying the retron DNA, though this is not shown) may trigger retron immunity while the glycosidase might have some other role which is not explored in any way.

Finally the authors used a LacZ-based assay to quantify the utility of their new retrons in bacterial genome editing compared to the previously characterized Eco1 retron. Confusingly, the authors did not use chromogenic LacZ substrates (e.g., "blue-white-selection" with X-Gal) but rather Illumina sequencing to quantify genome editing efficiency. Convincingly, the authors show that most new retrons have some detectable activity in genome editing though they are not as efficient as the Eco1 retron.

The discussion section is interesting and takes up some loose ends from this study, but it is (like the introduction) totally focused on the bacterial immunity function of the new retrons or their possible application in genome editing. Other relevant aspects - most importantly, specific limitations of this study - are not outlined or discussed.

Other points:

- future versions of the manuscript should have line numbers and page numbers to facilitate peer review

- S. quinivorans and H. chinensis are mentioned totally out of the blue and as an abbreviation (end of first paragraph of the results section) without having been introduced before with full name

- important things like controls, annotated size markers, etc. are missing from the polyacrylamide gels shown in this study (and the other experiments also have a quite minimalist setup)

- the manuscript uses a lot of jargon that is likely incomprehensible to non-experts if not explained ("PRTase", "NDT", etc.)

- the manuscript contains a variety of claims that I feel should be referenced but are not (e.g., "retrons often provide defense against phages when ported into a new species as well")

- unless I overlooked it, the paper does not mention that the ORFs in the retron loci have been codon-optimized when cloned for analysis in E. coli; this is obviously important and (if it has not been done) either has to be tested or mentioned as a critical limitation

- the figure captions and the annotations of content in the figures are often very minimalist and leave the reader a bit puzzled (e.g., Figure S1)

Reviewer #3 (Ilya Osterman):

Review of the Manuscript on Novel Retrons in Bacterial Immune Defense and Genome Editing

Overall Assessment:

The manuscript presents a well-structured and engaging study on newly discovered retrons and their roles in bacterial immunity and genome editing. The work is highly relevant to the fields of microbiology, bacterial immunity, and biotechnology. The study provides valuable insights into the diversity of retrons, their defensive potential, and their biotechnological applications. However, while the findings are intriguing, additional experiments are required to fully validate the claims, particularly in demonstrating the genome-editing potential of these retrons beyond E. coli and in expanding the analysis of their defensive capacity.

Major Concerns:

Genome Editing in Diverse Hosts:

The authors successfully demonstrate genome editing using these retrons in E. coli, which aligns with previous studies on retron-based genome engineering.

However, since the retrons were isolated from a variety of bacterial species, it would significantly strengthen the manuscript if the authors could show their potential for genome editing in their native hosts or in a broader range of bacterial species. For instance, demonstrating editing capabilities in Bacillus, Serratia, Pseudomonas, or other less commonly studied hosts mentioned in the text would enhance the novelty of the study.

Expanding the experimental scope to non-E. coli species would provide compelling evidence of the broader applicability of these retrons as genome-editing tools.

Phage Defense Potential:

The manuscript presents data on the ability of these retrons to protect E. coli against a panel of four phages.

While this is an important first step, testing these retrons against a larger panel of phages would provide more robust evidence of their defensive potential. A broader set of phages (e.g., 12 to 24 phages from well-curated collections such as the Basel Phage Collection) should be tested.

Additionally, it would be valuable to examine whether these retrons can provide phage defense in bacterial species other than E. coli, particularly their native hosts. This would help establish whether the protective effect is species-specific or generalizable.

---

## [Decision Letter · Decision Letter 2]

1 Oct 2025

Dear Seth,

Thank you for your patience while we considered your revised manuscript "Phage defense and genome editing using novel retrons sourced from isolated environmental bacteria" for publication as a Research Article at PLOS Biology. This revised version of your manuscript has been evaluated by the PLOS Biology editors, the Academic Editor and the original reviewers. I would like to mention that we all commend you for the the great work done in the revision.

Based on the reviews, we are likely to accept this manuscript for publication, provided you satisfactorily address the remaining points raised by Reviewer 1. Please also make sure to address the following data and other policy-related requests.

a) We routinely suggest changes to titles to ensure maximum accessibility for a broad, non-specialist readership, and to ensure they reflect the contents of the paper. In this case, we would suggest a minor edit to the title, as follows. Please ensure you change both the manuscript file and the online submission system, as they need to match for final acceptance:

"New retron systems from environmental bacteria identify triggers of anti-phage defense and expand tools for genome editing”

Please supply the numerical values either in the a supplementary file or as a permanent DOI’d deposition for the following figures:

Figure 3a-d, 4cfij, 5e, 6b-g

c) Please cite the location of the data clearly in all relevant main and supplementary Figure legends, e.g. “The data underlying this Figure can be found in S1 Data” or “The data underlying this Figure can be found in https://doi.org/10.5281/zenodo.XXXXX”

d) Thank you for providing the original, uncropped and minimally adjusted images supporting all blot and gel results. However, we required that all these are provided in a single PDF and name it "S1_raw_images2. You can read more about our guidelines for how to prepare and upload this data here: https://journals.plos.org/plosbiology/s/figures#loc-blot-and-gel-reporting-requirements

We expect to receive your revised manuscript within two weeks.

*Published Peer Review History*

*Press*

Sincerely,

Melissa

Melissa Vazquez Hernandez, Ph.D.

Associate Editor

PLOS Biology

REVIEWERS' COMMENTS

Reviewer #1:

The authors have done a terrific job revising this manuscript and adding exciting additional data. I especially commend them on digging into the Bth1 retron system to identify the locus and group II intron.

I have only very modest comments on the revision:

Line 272-274: I think there is a mistake in the description of this result i.e. I believe it is the WT strain with methyltransferase that shows reduced growth whereas the retron deleted strain is unaffected?

Supplementary files 1 and 2 were not included in the manuscript link sent to me so could not be reviewed.

Line 397: remove "and"?

Reviewer #2:

I thank the authors for the very comprehensive, thoughtful, and thorough revision of their work. This is now a fantastic manuscript that will find many interested readers. All my previous comments (including quite challenging ones) have been convincingly addressed, and I have no further comments.

---

## [Editor Report · Decision Letter 3]

7 Oct 2025

Dear Seth,

Thank you for the submission of your revised Research Article "New retron systems from environmental bacteria identify triggers of anti-phage defense and expand tools for genome editing" for publication in PLOS Biology. On behalf of my colleagues and the Academic Editor, Jeremy Barr, I am pleased to say that we can in principle accept your manuscript for publication, provided you address any remaining formatting and reporting issues. These will be detailed in an email you should receive within 2-3 business days from our colleagues in the journal operations team; no action is required from you until then. Please note that we will not be able to formally accept your manuscript and schedule it for publication until you have completed any requested changes.

PRESS

Sincerely, 

Melissa

Melissa Vazquez Hernandez, Ph.D., Ph.D.

Associate Editor

PLOS Biology
